# Analysis of Racing Greyhound Path Following Dynamics Using a Tracking System

**DOI:** 10.3390/ani11092687

**Published:** 2021-09-14

**Authors:** David Eager, Imam Hossain, Karlos Ishac, Scott Robins

**Affiliations:** 1Faculty of Engineering and Information Technology, University of Technology Sydney, P.O. Box 123, Broadway, NSW 2007, Australia; MDImam.Hossain@uts.edu.au (I.H.); Karlos.Ishac@uts.edu.au (K.I.); 2Greyhound Racing Victoria, 46-50 Chetwynd Street, West Melbourne, VIC 3003, Australia; srobins@grv.org.au

**Keywords:** greyhound racing track, IsoLynx tracking system, bio-loggers, clothoidal, Euler and Cornu spirals, velocity, acceleration, jerk, greyhound welfare, greyhound safety

## Abstract

**Simple Summary:**

The University of Technology Sydney (UTS) has been working collaboratively with the Australasian greyhound industry to reduce the frequency and severity of injuries. Where the UTS recommendations have been adopted, the injury rate has dropped significantly. This has been achieved by animal welfare interventions that lower racing congestion, and lower transient forces and jerk rates the greyhounds experience during a race. This study investigated the use of a greyhound location tracing system where small and lightweight signal emitting devices were placed inside a pocket in the jackets of racing greyhounds. The high magnitudes of velocity, acceleration and jerk posed significant technical challenges, as the greyhounds pushed the human tracking system beyond its original design limits. Clean race data gathered over a six-month period were analysed and presented for a typical 2-turn greyhound racing track. The data confirmed that on average, greyhounds ran along a path that resulted in the least energy wastage, which includes smooth non-linear paths that resemble easement curves at the transition between the straights to the semi-circular bends.

**Abstract:**

The University of Technology Sydney (UTS) has been working closely with the Australasian greyhound industry for more than 5 years to reduce greyhound race-related injuries. During this period, UTS has developed and deployed several different techniques including inertial measurement units, drones, high-frame-rate cameras, track geometric surveys, paw print analysis, track soil spring-force analysis, track maintenance data, race injury data, race computer simulation and modelling to assist in this task. During the period where the UTS recommendations have been adopted, the injury rate has dropped significantly. This has been achieved by animal welfare interventions that lower racing congestion, and lower transient forces and jerk rates the greyhounds experience during a race. This study investigated the use of a greyhound location tracing system where small and lightweight signal emitting devices were placed inside a pocket in the jackets of racing greyhounds. The system deployed an enhanced version of a player tracking system currently used to track the motion of human athletes. Greyhounds gallop at speeds of almost 20 m/s and are known to change their heading direction to exceed a yaw rate of 0.4 rad/s. The high magnitudes of velocity, acceleration and jerk posed significant technical challenges, as the greyhounds pushed the human tracking system beyond its original design limits. Clean race data gathered over a six-month period were analysed and presented for a typical 2-turn greyhound racing track. The data confirmed that on average, greyhounds ran along a path that resulted in the least energy wastage, which includes smooth non-linear paths that resemble easement curves at the transition between the straights to the semi-circular bends. This study also verified that the maximum jerk levels greyhounds experienced while racing were lower than the jerk levels that had been predicted with simulations and modelling for the track path. Furthermore, the results from this study show the possibility of such a systems deployment in data gathering in similar settings to greyhound racing such as thoroughbred and harness horse racing for understanding biomechanical kinematic performance.

## 1. Introduction

In Australasia, greyhound racing is a popular recreational sport. It is enjoyed by people of all ages from all around the country. Greyhound racing is conducted in all six states of Australia, the Northern Territory, New Zealand and several other countries. In Australasia, there are 59 greyhound racing tracks providing 199 different starts. The greyhound is an amazing animal having a maximum ground speed exceeding 20 m/s, making them twice as fast as the fastest human. They can accelerate to a maximum speed in less than 80 m in under 4 s. Greyhounds are elite athletes, and like elite human athletes, they sustain injuries performing the activity they love—galloping. Their gait is the highly efficient rotary gallop, which is shared with the Cheetah [1].

Typically, there are eight greyhounds in a race distance [2]. Races are conducted at purpose-built tracks and are run counter-clockwise. There are three broad categories of track, namely: straight, 1-turn and 2-turn. The 1-turn and 2-turn races are generally conducted on oval-shaped tracks with a mechanical lure system that the greyhounds chase [3]. Greyhound racing is conducted over three broad racing distance categories, namely:Short (sprint), up to 424 m;Medium (main), between 424 m and 569 m; andLong (staying), greater than 570 m.

This paper presents the results of a study that was conducted by the University of Technology Sydney (UTS) on racing greyhounds at a typical 2-turn greyhound racing track. The study obtained the x-y location of 6 months of races from a medium length (525 m) and two long length (600 m and 725 m) distances. The data obtained were from a location tracking system known as IsoLynx [4]. The data show each of the eight greyhounds’ real-time time stamped location coordinates together with the location coordinates of the mechanical lure. One of the major variables that indicates trajectory path smoothness is the rate of change in centrifugal acceleration (jerk) [5]. Furthermore, jerk is a parameter used for describing the experienced force profile on the body. It is an essential parameter for measuring safety when force is involved [6]. In [7,8], the authors explain jerk in the context of measuring the roller coaster safety performance and also give hints about its application in greyhound track path design. Using the data obtained, the greyhound kinematics such as speed and acceleration and path smoothness properties of the greyhound racing such as curvature and jerk were analysed.

The track chosen has physical features that are common to many 2-turn greyhound racing tracks around the world, namely:Two tight 52 m semi-circular bends (measured 1 m out from track inside);Two 40 m straights;Four short 20 m horizontal transitions between an 8% camber in and out of the bends and a 4% camber in and out of the straights;Minimal lateral alignment transition between the bends and the straights;A 600 m race distance, which starts into the bend section of the track (home turn); and525 m and 725 m race distances, which start into the straight sections of the track, namely the home straight and the back straight, respectively.

A plan view of the greyhound track used in this study is depicted in Figure 1. The plan on the left shows the location of the three starts at 525 m, 600 m and 725 m. The plan on the right is a plan showing key dimensions with the track path curvature combs shown in blue.

## 2. Background

UTS has been providing the Australasian greyhound industry with technical advice for more than 5 years in an effort to reduce race related injuries and improve the welfare of the greyhounds. Throughout this period, UTS has used several different devices and methods for analyses including inertial measurement units (IMU) [9,10,11,12], drone-based tracking, high-frame-rate (HFR) video capture [13], track survey data [14], paw print analysis [15], track soil spring-force analysis [16,17,18], track maintenance data [19], race computer simulation and modelling [20], and greyhound injury data [21].

Tracking systems have been widely researched and have been used to monitor both animals and humans [22,23,24,25,26,27]. These systems can be categorized based on the type of tracker and target tracking information. In this section, we present related works that we have used as a basis to guide our research in this paper.

In our research, we used the IsoLynx [4] system to track the greyhounds around a racing track in order to extract real-time information of their position on the track. This gives us insight into their racing behaviours and allows us to pinpoint areas on the track that may be of high risk. Previous researchers of animal tracking systems have implemented systems such as Global Positioning Systems (GPS), vision-based trackers, embedded IMU tracking and radio frequency locators.

Vision-based tracking is the most common tracking method for animal monitoring. The research in [28] tracks the motion of a cheetah using a camera attached to the body and an IMU and GPS sensor suite. Although this provides accurate information on the motion of the tail, the attachment of a bulky camera onto the cheetah encumbers it and affects its natural motion and behaviours. In tracking natural motion of animals, it is generally important to ensure that the attached device is as unobtrusive as possible so as to not affect the natural behaviours of the animal. The research in [29] used a background subtraction technique to track multiple animals at once, but the tracking is limited within the field of view of the system. This is feasible when the animals are restricted within an enclosed area. However, fixed vision-based trackers are not so suitable for outdoor use. Another problem with these systems is occlusions due to changing light and scene conditions, which also makes them difficult for outdoor use.

BioTracker is an open source visual animal tracker that provides the most common core functionalities for animal tracking [30]. It is especially intended to be compatible with low-cost hardware. However, it also faces the same limitations as the previous vision-based trackers. Most vision-based tracking systems for animals use a birds-eye view and are often fixed for indoor environments. Recently, vision-based systems have also been paired with AI algorithms and models to better estimate animal locations and behaviours. The research in [31] uses low-cost cameras fixed in an outdoor pasture in combination with a neural network algorithm to accurately track animals within a field of view. The neural network components assist with the problems of occlusion faced by many vision-based trackers by providing more intelligent insight through prediction models. The work in [32] tackles the field of view problem by implementing a freely moving camera for animal tracking. They have used a drone-based camera approach, which is capable of tracking animals outdoors and their trajectories. The research in [33] uses image recognition combined with AI-based algorithms to track animals within a zoo. This allows visitors to more conveniently locate the animal within its living area. The work in [34] used a neural network trained on prior racing data to attempt to predict future racing outcomes.

Beacon-based animal trackers have been used in the past and are popular for their allowed area of use. The work in [35] developed a batteryless beacon tracker for full-time tracking of sheep. The research in [36] presents the BATS system, an ultra low energy wireless sensor network for multiple animal tracking. It uses a radio frequency-based tracking system that makes use of wireless sensors attached to individual bats. These sensors communicate between each other and a receiver to provide information about the bats’ locations and flight patterns. Tracking animals underwater also has its challenges due to the limited number of compatible technologies. The work in [37] makes use of a combination of an IMU, depth sensor, GPS and Bluetooth Low Energy (BLE) to monitor animal behaviour underwater. The intended system is packaged in a sensor tag and attached to marine animals for wireless location sensing.

Similar tracking systems are also used for humans. Player tracking is commonly used in human sports to gain insight into location and dynamics. The research in [38] used GPS to track Chelsea soccer players to analyse player dynamics. The results showed that the centre forwards face the highest demand for high speed runs (HSR). The study conducted in [39] took advantage of known behaviours of jockeys in horse racing to tailor a tracking algorithm based on group dynamics. Our previous research [40] also analysed tracking Taekwondo punches through a combination of a camera and accelerometers. We used a high-speed-camera to track the motion of the practitioner and an accelerometer attached to a punching bag to measure the impact dynamics of the bag. We found this was optimal for this scenario, since it did not impede the practitioner’s motion.

IMU-based tracking of athletes has been especially common in high impact, quick motion sports such as martial arts. A systematic review of 36 papers presented in [41] highlights the key differences between inertial tracking systems for performance monitoring in combat sports. It was clear to see that research into these systems has advanced over the last few years, as there has been a need from professional organisations, such as the Olympics, to use these systems in martial arts for automatic classification of strikes, impact analysis and automatic scoring. Extensively, these systems have also been applied for coaching purposes. The research in [42] uses a biomechanical sensor to coach new practitioners in Wushu techniques.

Local positioning tracking systems have also recently been used to gain insight into player position and motion. The study in [43] used local position tracking on ice hockey athletes across five international games to determine differences in dynamics between forwards and defenders. Extensively, the system in [44] uses an experimental approach based on radio frequency BLE to track basketball players in an indoor environment. Although the study was not conducted on actual players, it shows promise for alternative indoor tracking applications. The research in [45] examined GPS tracking and survey data of Rugby Union athletes in order to enhance future injury prevention methods. The results showed that high-speed running in practice sessions was the leading contributing factor in managing future injuries among players.

Methods have also been explored for tracking the objects within sports. The research of Huang et al. [46] introduces a deep learning network called TrackNet for tracking small objects in sporting scenarios. The application presented analyses the motion of a tennis ball under problematic image capture scenarios. Similarly, the work in [47] uses embedded internet of things (IoT) sensors to track the 3D motion of a cricket ball in order to provide deeper insight for sports analytics. The study in [48] uses a combination of cameras and a convolutional neural network to track players and objects in a basketball game. It aims to use relative tracking between the player and ball to determine the overall percentage of ball possession automatically at the end of the game.

We have chosen to use IsoLynx for our greyhound tracking purposes because it provides accurate real-time tracking without impeding the animal’s natural behaviours. It is also robust and suitable for outdoor use under changing weather and lighting conditions.

## 3. Method

Greyhound real-time location coordinates data was gathered for 6 months of race starts for each race distance at the track. The data obtained was from greyhounds of various age groups and weights randomly picked from all the race meetings that occurred in the 6 months period. For the analyses, there were more than 1000, 250 and 16 data sets for 525 m, 600 m and 725 m respective race distances. The IsoLynx system recorded the X and Y coordinates and clock time for each greyhound from the time they leave the starting boxes to the finish line. The data sampling rate was 30 ± 1 samples per second. The IsoLynx system coordinates fluctuated when the carry-weight tags carried by the greyhounds were in close proximity to metals such as the starting boxes. Therefore, data was cleaned using a moving average filter where anomalies were identified and discarded. The trajectories of each greyhound start for each race distance at the track were generated where for the curvature analysis greyhound location coordinate data sampling rate was adjusted to match the average greyhound stride frequency of 3.34 Hz. Finally, only race start data sets with the full race duration were used for the analyses. For this reason, some analyses like trajectory analysis for the 725 m race distance starts were omitted. The average value of race parameters including greyhound speed, acceleration, path curvature and yaw rate was calculated by first finding the average in a race and then calculating the average of all races. This gave an overall picture of the race dynamics for a race distance. Figure 2 shows the procedure adopted to calculate race parameter values from the IsoLynx data. In the last step as shown in Figure 2, if outliers were present in the calculated race dynamics data, and also where data points deviated from three standard deviations, a moving average filter was applied to the calculated data in the calculation process queue. This made sure only the data sample which fell inside the true value sphere were taken into account for dynamics calculations.

Figure 3 contains two photographs of the IsoLynx transponder poles deployed at a typical greyhound track.

Figure 4 depicts the IsoLynx greyhound location tracking system deployment at a track. The system consisted of reference nodes or transponders, back-end computers or servers and tags for the greyhounds (greyhound transponder). The IsoLynx is a proprietary system where the reference nodes or transponders around the track periodically interact with the tag carried by the greyhound to achieve a location accuracy of ±150 mm. The real-time location data are then transmitted to the server computer, where they are stored.

### System Limitations

While processing the greyhound coordinate data obtained from the IsoLynx system, a small portion of data samples for some race starts deviated from the true values and were discarded and replaced with moving average values. For instance, Figure 5 shows the X coordinates data samples for a single greyhound, where data points which deviated from the true values were replaced by applying moving average filter values to the data samples. Furthermore, race data which cannot be considered accurate, such as the initial couple of hundred milliseconds into the race, were not included in the analyses. This is because the IsoLynx system could not gather reliable data for this initial period into the race due to the metal starting boxes physically interfering with the greyhound carry tag signals.

## 4. Results and Discussion

The results presented in the following sections can be broken down based on where they occurred at the track.

For 525 m distance starts, the first bend (back turn) commenced approximately 71 m from the start and ends at approximately 249 m. This was followed by back straight, which was located between approximately 249 m and 309 m from the start. The second bend (home turn) was located between the 309 m and 462 m, followed by the home straight to the finish line.

For the 600 m starts, the first bend (back turn) commenced approximately 147 m from the start and ends at approximately 323 m. This was followed by back straight, which was located between approximately 323 m and 377 m from the start. The second bend (home turn) was located between 377 m and 538 m, followed by the home straight to the finish line.

For 725 m distance starts, the first bend (home turn) commenced approximately 54 m from the start and ends at approximately 219 m. This was followed by home straight, which was located between approximately 219 m and 274 m from the start. The second bend (back turn) was located between the 274 m and 452 m, followed by the back straight, which ends at approximately 503 m from the start. The final bend, which followed back straight ends at approximately 671 m, followed by the home straight to the finish line.

### 4.1. IsoLynx Location Coordinates Data

Figure 6 is a screen grab (https://bit.ly/3iDd6Sa (accessed on 21 July 2021)) of race data visualisation for a single race event from a 525 m start distance. The video shows the race data being projected in the track 2D model as it was recorded by the IsoLynx system. It depicts the eight greyhounds with their individually coloured jackets chasing the lure, which is shown as a small orange coloured oval. The dynamic data calculated from the location coordinates of the racing subjects are presented at the top of the video and include: the overall race time; the average speed of the greyhounds; the lure distance travelled; the distance between the lead greyhound and the lure; and the relative speed between the lure and the lead greyhound. In the video, the greyhound race track model was coloured according to different virtual track lanes to observe the veering of the greyhounds between the lanes.

### 4.2. Greyhound Running Trajectory

As the greyhounds ran around the track, they took specific paths to complete the race distance. To find the overall path trajectory of all the greyhounds, the coordinate data of all greyhounds starts were averaged. Figure 7 depicts the average trajectory of all greyhounds for 525 m race distance starts where the blue line represents the track shape and the dotted red line denotes the average greyhound running trajectory. It is evident that the average of the greyhound running paths differs from the track path where it approximates an oval shape with a high degree of path smoothness between the straights and the bends. A similar result was observed for the 600 m race distance starts.

From the track survey data, the inside lure rail coordinate data was extracted to calculate the greyhound’s perpendicular distance to the lure rail or track inside boundary. Figure 8 shows the average perpendicular distance to the track inside boundary of all racing greyhound starts for both the 525 m and 600 m racing distances. As can be seen from the plot, no distance data are presented for up to 2.5 s or 25 m into the race. This is because, for this portion of the race, the data were not sufficiently reliable to calculate greyhound proximity to the rail. The plot shows that the greyhound perpendicular distance from the track inside boundary varied and fluctuated over the race distance for both the 525 m and 600 m distance starts. It can be seen that in the straight sections of the track, the greyhound offset distance from the track inside boundary was significantly higher than in the bend sections where the greyhound perpendicular distance to the rail was just under 1 m.

For instance, for the 525 m start, between 245 m and 300 m, greyhounds traversing through the track back straight averaged a perpendicular distance to the rail which peaked at just above 1.5 m. Likewise, for the 600 m start, between 375 m and 535 m, greyhounds traversing through the track home-turn bend averaged a perpendicular distance to the rail of around 0.5 m. Furthermore, from the plots, it was seen that for both the 525 m and 600 m distance races that with distance into the race greyhounds got closer to the track inside rail as the lower proximity-to-the-rail values indicate. Finally, as shown in the plots, the data showed that for the 600 m race starts, greyhounds were closer to the rail in the track bend sections compared to the 525 m race starts.

### 4.3. Greyhound Speed Profile

From the race location coordinates and clock data, the instantaneous speed data of all greyhound starts were calculated. By taking the average of all instantaneous speed data the average greyhound speed profile was calculated. The average greyhound racing speed profiles for the 525 m, 600 m and 725 m racing distance starts are presented in Figure 9. It can be seen that a typical greyhound’s racing speed rapidly increased to a maximum speed slightly below 19.5 m/s at between 80 m and 150 m into a race. For medium length race distances (525 m and 600 m), the greyhound’s speed gradually reduced until the end of a race, where it was approximately 16.1 m/s. For the longer race distance (725 m) the greyhound’s speed gradually reduced until the end of a race where it was below 15.5 m/s. Furthermore, it can be observed that for 600 m starts, which were located at the track bend, the peak speed was highest of all race distance starts, while for this distance start, the initial acceleration phase of the greyhounds was longest among all the race distance starts.

### 4.4. Curvature of Greyhound Running Path

The curvature of greyhound running paths was calculated by finding the inverse of the radius of curvature of greyhound turns at different points in time from the race data by using the formula for the circumradius of a triangle. Curvature data showed path smoothing properties of the greyhound running path for both the 525 m and 600 m racing distance starts. Furthermore, as the curvature is proportional to the radius of turn it was used for identifying the greyhound turning radius on the straight section of the track. Figure 10 shows the average curvature of greyhound running paths for both the 525 m and 600 m race distance starts. The average curvature value was highest in the bend sections of the track at just under 0.0197 1/m and lowest in the straight track sections at just under 0.004 1/m. This translates to approximately 50.8 m and 250 m greyhound turning radii at the bend and the straight, respectively. As evident from the greyhound trajectory plot, the average running path of the greyhounds approached a round path throughout the races where the curvature was greater than zero for the distance of the race. Furthermore, average curvature plots show for all race distance starts that there was a rapid decrease in the curvature while exiting the first bend at around 190 m and 265 m distances for the 525 m and 600 m starts, respectively. This indicates a larger radius smooth path exit from the bend by the greyhounds. However, the same cannot be said for the second track bends. In Figure 10, the average curvature plot of greyhound running paths approaches a non-linear curve where the curvature value changes gradually and the greyhound turning radius can be seen to be smooth as found in the running path with a higher degree of path smoothing properties. Finally, for 525 m distance starts the lowest and highest values of average curvature in the track bend were 0.0189 1/m and 0.0195 1/m, respectively. The corresponding lowest and highest values of average turning radii were 52.9 m and 51.3 m for 600 m distance starts, and the lowest and highest values of average curvature in the track bend were 0.0189 1/m and 0.0197 1/m, respectively. The corresponding lowest and highest values of average turning radii were 52.9 m and 50.8 m.

### 4.5. Yaw Rate of Greyhound Heading

The greyhound heading yaw rate was found from the greyhound instantaneous speed and turning radius, where the average yaw rate of greyhound heading was calculated and indicated how quickly greyhounds were changing their direction on the track. Figure 11 shows the greyhounds’ average yaw rate for both the 525 m and 600 m race distance starts. In the track bend sections, the greyhounds’ average yaw rate was higher after entering the bend and lower as they approached near the end of the bend. A negative average yaw rate was observed as the greyhounds came out from the boxes for the 525 m starts between 3.0 s and 4.8 s into the race indicating the clockwise direction of greyhound heading. The highest average yaw rate was just under 0.35 rad/s, which occurred after entering the first bend at around 147 m and 225 m distances for the 525 m and 600 m distance race starts, respectively.

### 4.6. Greyhound Acceleration Profile

The greyhound forward race acceleration was calculated using the instantaneous speed values. Figure 12 depicts average forward acceleration of greyhounds for the 525 m, 600 m and 725 m racing distance starts. It should be noted that the acceleration between zero and one second into the race could not be calculated as the data were deemed unusable. From the average forward acceleration plot, it can be seen that greyhounds accelerated rapidly out of the boxes reaching a maximum forward acceleration beyond 4.0 m/s${}^{2}$ at approximately 2 s after the race start. Afterwards, the average greyhound forward acceleration decreased sharply until it fluctuated between 0.26 m/s${}^{2}$ and −0.5 m/s${}^{2}$.

From the greyhound running path curvature and the instantaneous speed of the greyhound the centrifugal acceleration $a_{c}$ of the greyhound running path was calculated using Equation (Equation 1) where *s* denotes instantaneous speed of the greyhound and κ denotes greyhound running path curvature value. Figure 13 depicts average centrifugal acceleration of greyhounds for the 525 m, 600 m and 725 m racing distance starts. The average centrifugal acceleration was highest for 600 m starts at approximately 6.2 m/s${}^{2}$, which occurred after entering the first bend at around the 205 m distance.
(1)$$a_{c}=s^{2}\kappa$$

Figure 14 shows the vector sum of average greyhound forward and centrifugal acceleration showing the overall average acceleration experienced by the greyhounds for the 525 m, 600 m and 725 m racing distance starts. As can be seen from the plot for the 600 m starts, the period of initial high forward acceleration coincides with centrifugal acceleration making this distance the longest-lasting high acceleration distance start of all race distance starts.

### 4.7. Racing Greyhound Jerk Level

By using greyhound path curvature and instantaneous speed derived centrifugal acceleration data averages, minimum, and maximum jerk levels for the greyhounds running path were calculated using Equation (Equation 2). In Equation (Equation 2), the greyhound stride period was determined by assuming a constant greyhound stride frequency of 3.54 Hz. Furthermore, the theoretical jerk level for the track path following was modelled for comparison with the greyhounds running path average, minimum and maximum jerk levels. Figure 15 and Figure 16 compare greyhound running path averages, minimum and maximum centrifugal acceleration jerk to theoretical track path jerk level for the 525 m and 600 m race distance starts at the track. As can be seen from the plots, greyhound running path jerk levels were significantly lower than that of the jerk level to which the greyhounds are exposed by the track path. The maximum theoretical track path jerk level for approximately 1 m out from the track inside lure rail is approximately 8.0 m/s${}^{3}$ and 9.0 m/s${}^{3}$ for the 525 m and 600 m race distance starts, respectively. The maximum jerk level for greyhound running paths was just under 3.0 m/s${}^{3}$ for both the 525 m and 600 m race distance starts. This shows that the greyhounds’ running path minimised the jerk level from the physical track path.
(2)$$jerk=\frac{centrifugal\;acceleration\;in\;next\;stride-centrifugal\;acceleration}{stride\;period}$$

## 5. Conclusions

This study investigated the application of location tracking data in the context of finding racing greyhound path dynamics. The data presented in this paper show racing greyhound running paths where greyhounds adapted to a unique running path different from the track path. By analysing different parameters derived from the location tracking data greyhound motion and running trajectory were verified for different race distance starts. Greyhound running path trajectory was analysed by looking into its dynamic effect on the greyhound and also compared to the track shape. The results presented in this paper indicate that greyhounds optimised their running path where they followed a smooth non-linear trajectory.

From our research, we can draw several conclusions about greyhound racing dynamics. We observed that a typical greyhound’s racing speed rapidly increased to a maximum speed slightly below 19.5 m/s at between 80 m and 150 m into a race. For medium length race distances (525 m and 600 m) the greyhound’s speed gradually reduced until the end of a race where it was approximately 16.1 m/s. For longer race distance (725 m) the greyhound’s speed gradually reduced until the end of a race where it was below 15.5 m/s. The average curvature value was highest in the bend sections of the track where turning radii were at just under 50.8 m and 250 m at the bend and the straight, respectively. The curvature results also show that the curvature profile of the greyhound path following inside the bend is not exactly identical between the track bends (home and back turns). The curvature profile of the greyhound in the first bend deviated more from the track path than the second bend when it is assumed both bends are exactly identical in design and shape. This showed that the influence of the first bend was more significant in the greyhound path following than the second bend. As evident from the greyhound trajectory plots, the average running path of the greyhounds approached a round path throughout the races where the curvature was greater than zero for the distance of the race.

The highest average yaw rate observed was just under 0.35 rad/s, which occurred after entering the first bend at around 147 m and 225 m distances for the 525 m and 600 m distance race starts, respectively. The average centrifugal acceleration was highest for 600 m starts at approximately 6.2 m/s${}^{2}$, which occurred after entering the first bend at around the 205 m distance. The maximum jerk level for the greyhound running path was just under 3.0 m/s${}^{3}$ for both the 525 m and 600 m race distance starts. This shows that the greyhound’s running path minimised the jerk level from the physical track path.

As indicated from the disparity of jerk levels between the greyhounds trajectory and track path a greyhound dynamics compatible track path can be designed by formatting the sections of the track where it would optimise the force profiles acting on the greyhounds. For instance, railway tracks and highway road designs utilise transition curves such as spiral easements between the straight and bend sections to optimise efficiency and minimise unwanted force profiles.

The results from this study show the possibility of such a systems deployment in data gathering in similar settings to greyhound racing such as thoroughbred and harness horse racing for understanding biomechanical kinematic performance.

Our research provides fundamental insight into greyhound motion behaviours and sets the foundation for further work in this field to be explored. By gaining a deeper understanding of greyhound motion and behaviour we may be able to further minimise the number of injuries in greyhound racing and support healthier lives for the greyhounds. Extended analysis of future research based on this paper will also help for designing safer tracks for greyhounds.

## Figures and Tables

**Figure 1 animals-11-02687-f001:**
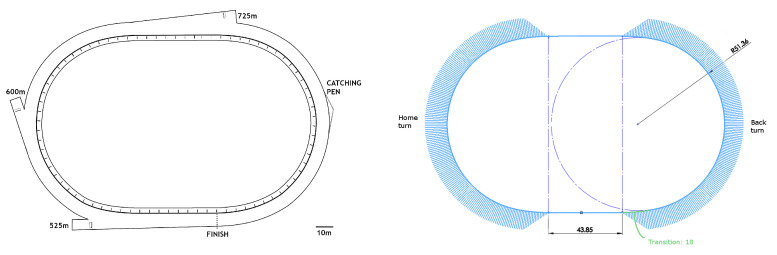
A plan view (**Left**) and dimension plot with curvature combs (**Right**) of the 2-turn greyhound track with 525 m, 600 m and 725 m racing distances.

**Figure 2 animals-11-02687-f002:**
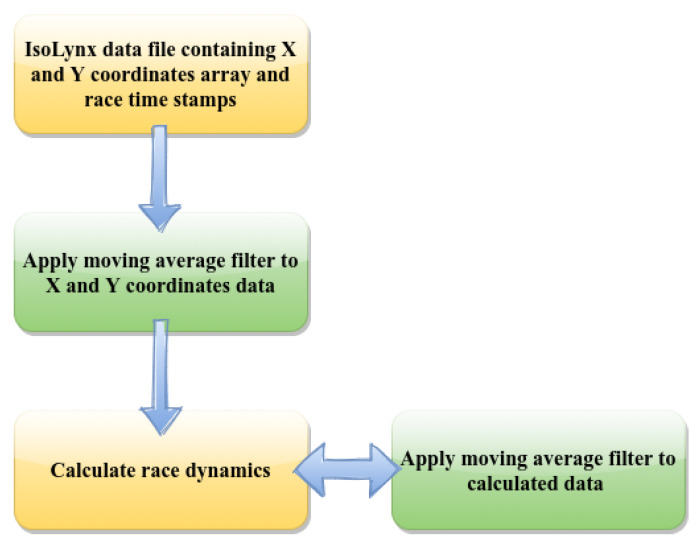
Steps followed to calculate the race parameter values from IsoLynx data.

**Figure 3 animals-11-02687-f003:**
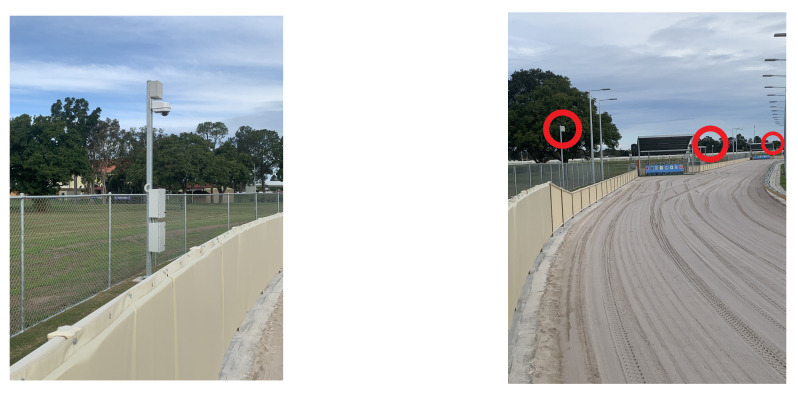
IsoLynx transponder poles. (**Left**) transponder and CCTV mounted on the top of a pole for a typical greyhound race track. (**Right**) three transponders (red circles) mounted along the back straight of a typical greyhound race track.

**Figure 4 animals-11-02687-f004:**
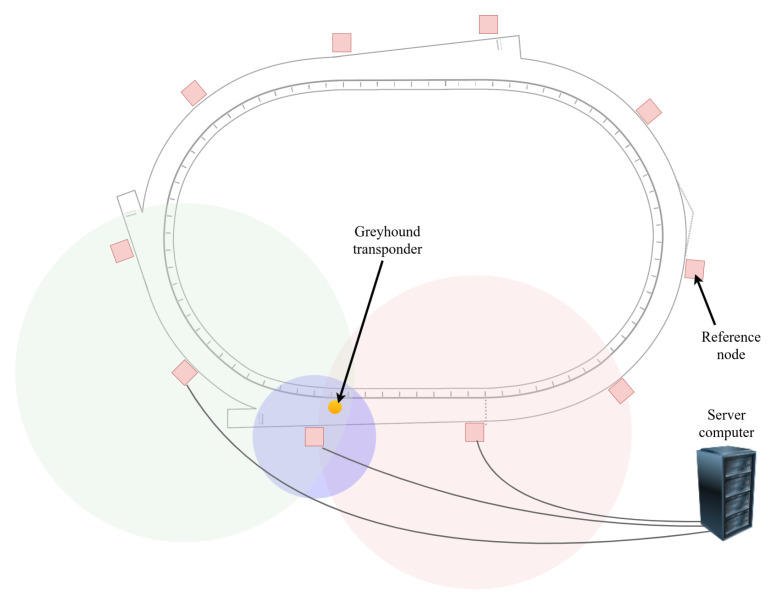
Diagram of IsoLynx system setup for a greyhound track.

**Figure 5 animals-11-02687-f005:**
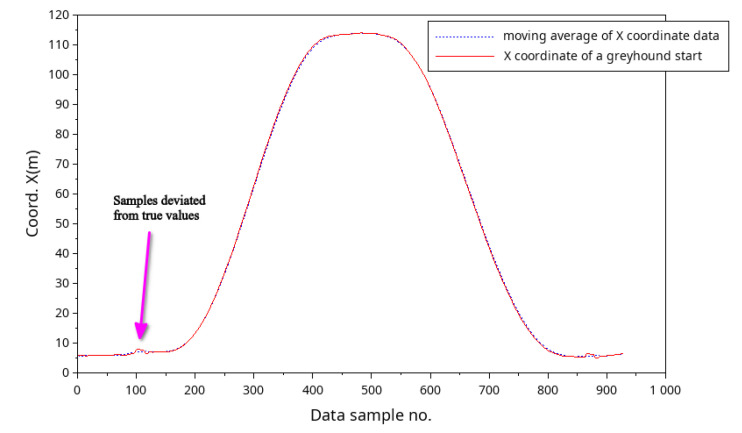
A moving average filter was used to correct data sample deviation from the true values.

**Figure 6 animals-11-02687-f006:**
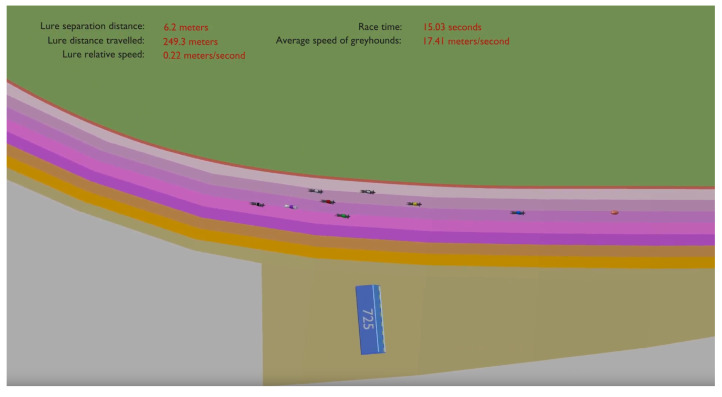
Screen grab (https://bit.ly/3iDd6Sa (accessed on 21 July 2021)) of the 525 m start race data replay for a single race as the greyhounds pass the 725 m starting boxes and enter the home straight of the track.

**Figure 7 animals-11-02687-f007:**
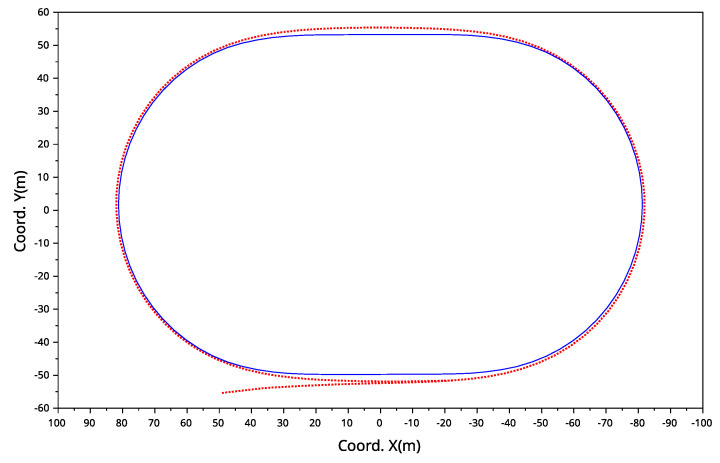
Average running trajectory for the 525 m starts where the greyhounds average running path is depicted by the dotted red line and the track shape (track inside boundary) by the blue line.

**Figure 8 animals-11-02687-f008:**
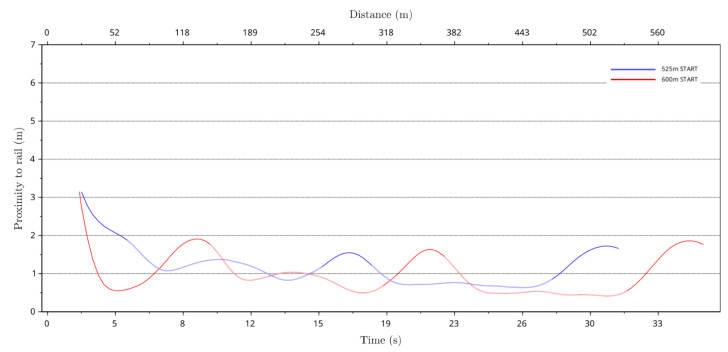
Average perpendicular distance of racing greyhounds from the track inside lure rail versus race time and race distance for both the 525 m and 600 m distances race starts. The lighter colours in the plots indicate the turn sections of the track.

**Figure 9 animals-11-02687-f009:**
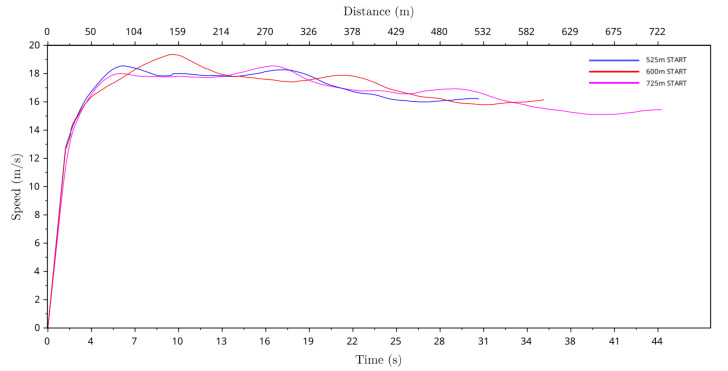
Average speed of greyhounds versus race time and distance for the 525 m, 600 m and 725 m racing distance starts. Distance and time commence (0 m and 0 s) at the respective boxes which were positioned around the circumference of the track as shown in Figure 1.

**Figure 10 animals-11-02687-f010:**
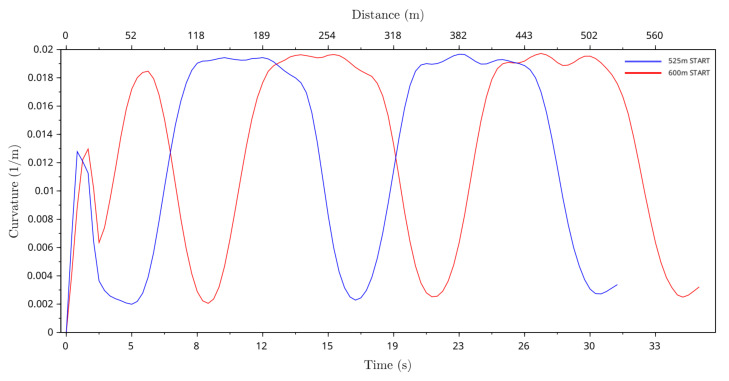
Average curvature of the greyhound running paths versus race time and race distance for 525 m and 600 m race distance starts.

**Figure 11 animals-11-02687-f011:**
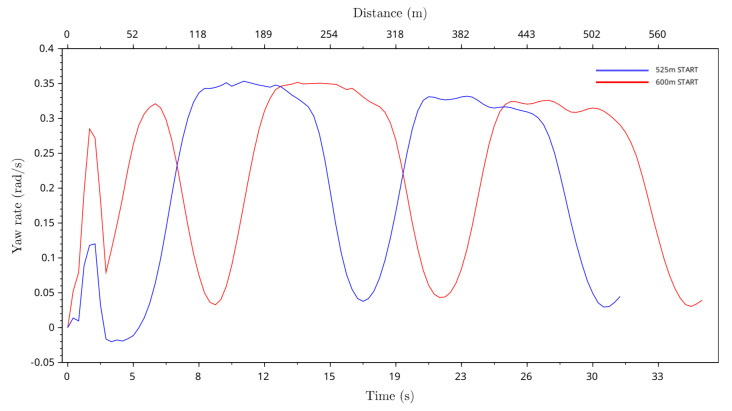
Average yaw rate of the greyhounds running path versus race time and race distance for 525 m and 600 m race distance starts.

**Figure 12 animals-11-02687-f012:**
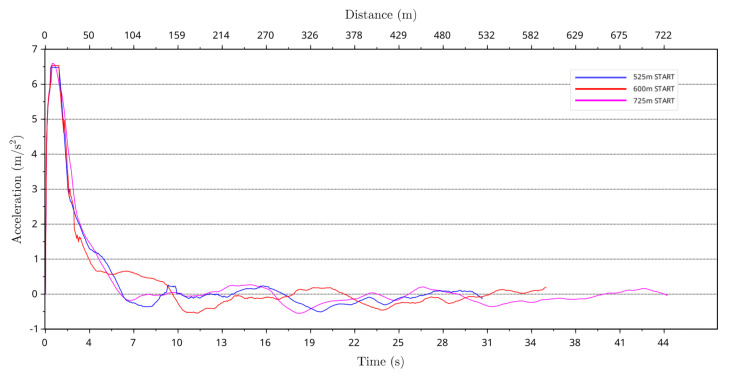
Average forward acceleration of greyhounds versus race time and distance for the 525 m, 600 m and 725 m racing distance starts.

**Figure 13 animals-11-02687-f013:**
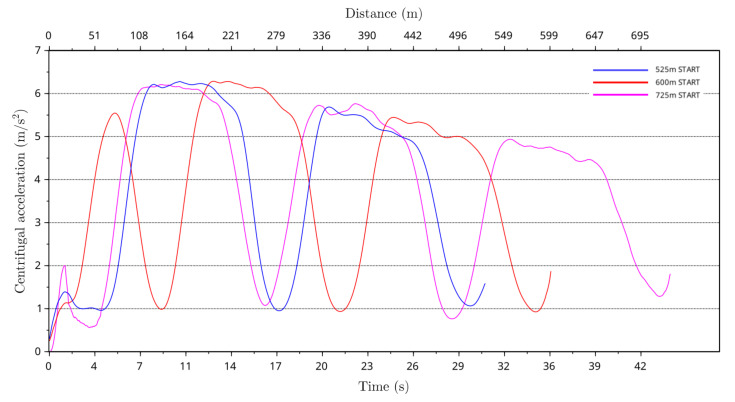
Average centrifugal acceleration of greyhounds versus race time and distance for the 525 m, 600 m and 725 m racing distances starts.

**Figure 14 animals-11-02687-f014:**
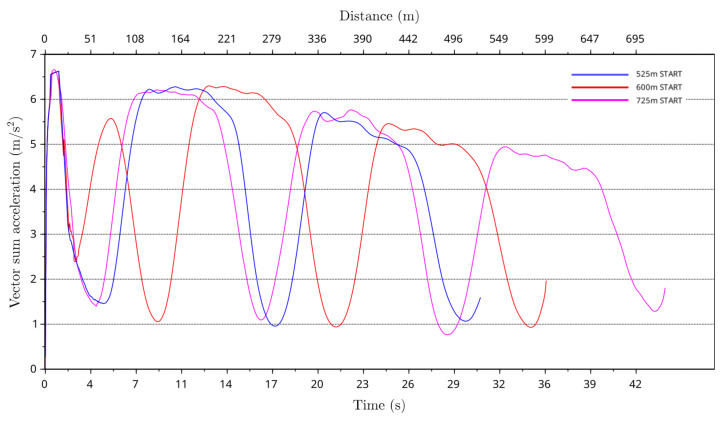
Total acceleration of greyhounds versus race time and distance for the 525 m, 600 m and 725 m racing distances starts.

**Figure 15 animals-11-02687-f015:**
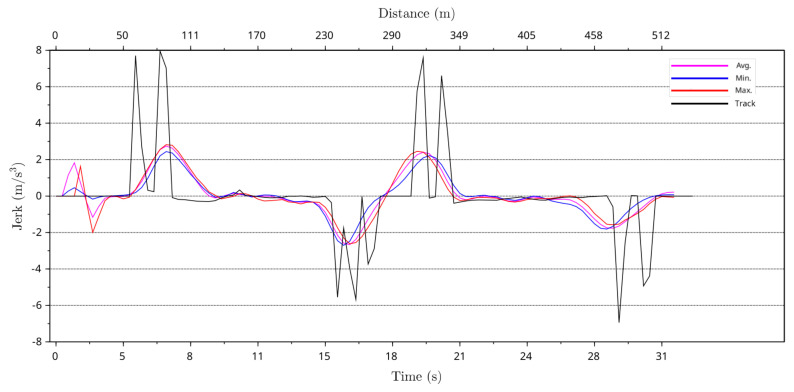
Maximum, minimum and average jerk of greyhounds running paths versus race time and race distance for 525 m distance race starts with a black plot showing the jerk as predicted from the shape of the lure rail.

**Figure 16 animals-11-02687-f016:**
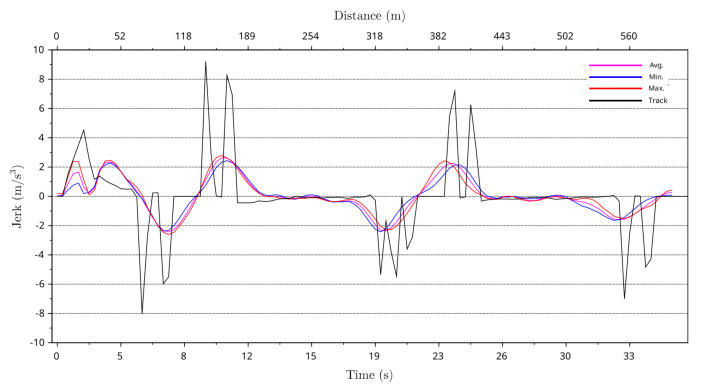
Maximum, minimum and average jerk of greyhound running paths versus race time and race distance for 600 m distance race starts with a black plot showing the jerk as predicted from the shape of the lure rail.

## Data Availability

Restrictions apply to the availability of these data. The data are not publicly available due to restrictions within the research agreement.

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
