# Peer review of "Analysis of Racing Greyhound Path Following Dynamics Using a Tracking System"

_animals, 2021, doi:10.3390/ani11092687_

Round 1
Reviewer 1 Report
This is a polished manuscript and I have few comments. The annotated PDF attached contains some markup corrections and some highlighted sections with notes. The notes are copied below in case they don't show up.
Figure 8 - Is there any way to mark on the plots where the turns are?
P12 "jerk" - Could this be explained in the Introduction?
Conclusions - Were the bends identical?
Should tracks be changed to better support a running path that minimises jerk level?

Author Response
File

Reviewer 2 Report
The manuscript is a well written article analyzing the path following dynamics of racing greyhounds in a 2-turn track model.
The methods section needs clarification on overall data size. There are eight dogs run at a given time for a given race and all races are recorded over a 6 month period. However, it is not clear how many individual dogs this data represents. Are the same eight dogs repeatedly used or are these multiple sets of eight dogs representing more than eight individuals across the data presented? Additionally, though the collection window for data spanned 6 months, what are the general number of races of each (525m, 600m, and 725m) that are represented across that period of time?
The calculations, such as centrifugal acceleration and jerk, would benefit from showing the equations used.
Minor grammatical and formatting corrections needed in line 24 (omit "of"), line 38 (correct letter for number "t0"), line367 (omit repeated "several"), Figure 3. (orient both images in portrait), Figure 10. (add 600m to caption).
Author Response
File
